# Hypoxia-Regulated Proteins: Expression in Endometrial Cancer and Their Association with Clinicopathologic Features

**DOI:** 10.3390/diagnostics14161735

**Published:** 2024-08-09

**Authors:** Dae Hyun Song, Jae Yoon Jo, Cho Hee Kim, Min Hye Kim, In Ae Cho, Jeong Kyu Shin, Won Jun Choi, Jong Chul Baek

**Affiliations:** 1Department of Pathology, Gyeongsang National University School of Medicine, Gyeongsang National University Changwon Hospital, 11, Changwon-si 51472, Republic of Korea; daehyun@gnu.ac.kr; 2Institute of Medical Science, Gyeongsang National University, Jinju 52727, Republic of Korea; poodeeng@naver.com (J.Y.J.); chohing9@gmail.com (C.H.K.); joymine86@naver.com (M.H.K.); obgychoia@gnu.ac.kr (I.A.C.); 2848049@hanmail.net (J.K.S.); choiwj@gnu.ac.kr (W.J.C.); 3Department of Obstetrics and Gynecology, Gyeongsang National University School of Medicine, Gyeongsang National University Hospital, Jinju 52727, Republic of Korea; 4Department of Pathology, Gyeongsang National University School of Medicine, Gyeongsang National University Hospital, Jinju 52727, Republic of Korea; 5Department of Obstetrics and Gynecology, Gyeongsang National University School of Medicine, Gyeongsang National University Changwon Hospital, 11, Changwon-si 51472, Republic of Korea

**Keywords:** endometrial cancer, HIF-1α, GLUT-1, tumor microenvironment, prognostic factors

## Abstract

Background: Hypoxia-regulated proteins (HIF-1α and GLUT-1) have been identified as prognostic markers in various cancers; however, their role in endometrial cancer remains unclear. This study aimed to evaluate HIF-1α and GLUT-1 expression in endometrial cancer and correlate their expression with clinicopathological features. Materials and Methods: A tissue microarray (TMA) was constructed using specimens from a retrospective cohort of 51 endometrial cancer patients who underwent hysterectomy at the Gyeongsang National University Hospital between 2002 and 2009. Clinicopathologic data were collected from electronic medical records, and HIF-1α and GLUT-1 expressions were assessed in the tumor tissue. Results: GLUT-1 expression in endometrial cancer was categorized as mosaic, central, or diffuse. Most patients (56.0%) exhibited a central pattern, followed by diffuse (32.0%) and mosaic (12.0%) patterns. GLUT-1 expression was not significantly associated with histologic grade (*p* = 0.365). HIF-1α expression in immune cells, but not tumor cells, was significantly associated with a higher histologic grade. A higher proportion of HIF-1α-positive immune cells, using both thresholds (≤1% vs. >1% and ≤5% vs. >5%), was significantly associated with higher histologic grade (*p* = 0.032 and *p* = 0.048, respectively). In addition, a higher proportion of HIF-1α-positive immune cells was significantly associated with a diffuse GLUT-1 expression pattern using >5% as a threshold. There were no significant differences in the proportion of HIF-1α-positive immune cells between groups stratified by age, tumor size, or invasion depth, regardless of whether the 1% or 5% threshold for HIF-1α positivity was used. Conclusions: A higher proportion of HIF-1α-positive immune cells is associated with endometrial cancers with higher histologic grade and diffuse GLUT1 expression patterns. These findings suggest a potential role for HIF-1α as a prognostic marker and highlight the need for further studies into the role of HIF-1α in the tumor microenvironment.

## 1. Introduction

Endometrial cancer is the sixth most commonly diagnosed female genital organ cancer in women, and its incidence has increased worldwide. It is the second most common gynecologic cancer in the world after cervical cancer [1]. The incidence of endometrial cancer has steadily increased in South Korea. In 2020, it surpassed the incidence of cervical cancer as the most common female genital tract cancer and exhibits a similar pattern to that of Western countries [2].

HIF-1α is a transcription factor that has an important role in enabling cells to adapt to hypoxic environments. HIF-1α is involved in various biological processes, particularly those associated with various cancer processes, including cell proliferation, apoptosis, angiogenesis, glucose metabolism, and energy metabolism [3]. As tumors grow, they require a continuous supply of nutrients and oxygen, which is achieved through neovascularization. This process not only facilitates tumor growth and expansion but also promotes metastasis [4]. The disorganized and uncontrolled proliferation of cancer cells results in an abnormal vasculature with chaotic architecture, increased blood flow, and enhanced vascular permeability. Consequently, the essential oxygen and nutrients required for tumor metabolism become deficient [5]. Thus, the development of hypoxia is a common occurrence in many solid tumors.

Tumor cells, unlike normal cells, exhibit elevated glucose metabolism, resulting in marked metabolic activity differences to maintain homeostasis. Normal cells derive approximately 90% of their energy from mitochondrial respiration, whereas less efficient glycolysis contributes to approximately 10% of the energy production [6]. In contrast, within the oxygen-restricted tumor microenvironment, cells adapt to a limited oxygen supply by generating the majority of their energy through glycolysis. This enables tumor cells to survive under hypoxic conditions as described previously [7]. The largest group of functionally regulated genes under the control of HIF-1α under hypoxic conditions is related to glucose metabolism [8,9]. Among these functions, HIF-1α enhances glucose uptake into cells by inducing the expression of facilitative glucose transporters, GLUT-1 and GLUT-3 [9,10]. GLUT-1 and GLUT-3, members of the transmembrane protein family and among the 14 known GLUT isoforms, were the first to be discovered [11]. They are expressed at varying levels in different human tissues. GLUT-1 is expressed in erythrocytes, germinative cells, and endothelial cells of the blood–brain barrier. Although it does not constitute a significant proportion of GLUTs in normal cells, its overexpression has been observed in colon, lung, and ovarian cancer as well as squamous cell carcinoma of the head and neck [12,13]. The proliferation and growth of solid tumors are characterized by increased glucose uptake and utilization rates, accompanied by the upregulated expression of various abnormal GLUT isoforms [14,15]. In head and neck cancers and acute myeloid leukemia, increased HIF-1α levels along with increased GLUT expression are associated with poor prognoses, such as advanced-stage disease and chemoresistance [16,17].

However, in the context of endometrial cancer, the prognostic and predictive value of HIF-1α remains unclear because of inconsistent findings across studies and limited research examining its correlation with specific clinicopathological features [18,19,20]. In the hypoxic tumor microenvironment, HIF-1α upregulates GLUT-1 expression, and the observed positive correlation between HIF-1α and GLUT-1 expression suggests potential crosstalk between these proteins [16,17,20,21]. This study aimed to examine the correlation between HIF-1α and GLUT-1 expression, tumor histologic grade, and clinicopathological features in patients with endometrial cancer.

## 2. Material and Methods

### 2.1. Sample Collection

The material for this study was sourced from samples collected at the Department of Pathology, Gyeongsang National University Hospital in Jinju City, South Korea, between January 2002 and December 2009. We procured 51 formalin-fixed, paraffin-embedded tissue block samples from patients who were diagnosed with endometrioid adenocarcinoma. All patients underwent a total abdominal hysterectomy with bilateral salpingo-oophorectomy, pelvic lymph node dissection, and omentectomy performed by an experienced gynecological oncologist. The diagnosis was confirmed by two pathologists. Disease staging and the histological type and grade of the tumor were determined according to the International Federation of Gynecology and Obstetrics (FIGO) and tumor-node-metastasis (TNM)-based Union for International Cancer Control (UICC) criteria [22,23]. The clinical data, including demographic and pathologic data for the cohort, were reviewed retrospectively with patient electronic medical charts. Informed consent was obtained from all patients, and the study was approved by the Institutional Review Board of Gyeongsang National University Hospital (GNUH-2022-03-009).

### 2.2. TMA and Immunohistochemical Analysis

Immunohistochemical staining was performed on tissue microarray (TMA) paraffin blocks. A monoclonal antibody against HIF-1α (1:100 dilution, ab51608; Abcam, Cambridge, MA, USA) and an anti-monoclonal antibody to GLUT-1 (1:300 dilution, ab11570; Abcam, Cambridge, MA, USA) were used as primary antibodies. The pattern of immunohistochemical staining was evaluated by two pathologists. Of the 51 samples, a total of 50 samples were readable on the GLUT-1 slides, and non-tumorous tissue was observed in 4 of 50 samples in the GLUT-1 slides. For the HIF-1α immunostaining slides, 49 tissue samples were readable, and non-tumorous tissue was observed in 3 of 49 samples. GLUT-1 was considered positive if it was strongly stained in the cytoplasm and cell membrane of cancer cells and classified into a central, mosaic, or diffuse pattern based on the distribution of GLUT-1 in the positive cells (Figure 1). In normal endometrial tissue, there was no evidence of GLUT-1-positive cells, indicating that the expression of GLUT1 is minimal or absent in cells of the normal endometrium. For the mosaic pattern, GLUT-1-positive cells were observed in discrete spots within the tissue. This suggests a spatial heterogeneity of GLUT1 expression within the tissue. For the central pattern, GLUT-1-positive cells were mainly concentrated around the lumen of the gland, suggesting that these cells, which line the glandular structures, require a high level of glucose uptake. For the diffuse pattern, cells that were strongly positive for GLUT-1 were evenly distributed throughout the tissue, suggesting a more uniform distribution of GLUT1 expression throughout the tissue section.

HIF-1α was moderately stained in the cytoplasm of cancer cells and tended to be co-expressed in GLUT-1-positive cells. A pattern of strong positive staining in the cytoplasm of immune cells rather than cancer cells was observed. The percentage of HIF-1α-positive immune cells in the tissue cores was measured and recorded by two pathologists. The percentage thresholds were 1% and 5%, respectively. To select patients for immunotherapy, tumor tissue is immunohistochemically stained for PD-L1 or PD-1 and the proportion of positive immune cells is assessed by pathologists using the 1% cut-off. Some studies have suggested a 5% cut-off. We borrowed the PD-1 and PD-L1 reading standards from the HIF-1α immune cell measuring system. The 1% and 5% thresholds used in this paper with the HIF-1α immune cell measuring system refer to the PD-1 and PD-L1 detection criteria [24,25]. Figure 2 shows that HIF-1α and GLUT-1 were co-expressed in the same region of the tissue samples. This suggests a potential interaction between these proteins and a concerted role in the tumor microenvironment.

### 2.3. Statistical Analysis

Association analyses of categorical variables were performed using the chi-squared test and Fisher’s exact test to determine the association between HIF-1α, GLUT-1 expression, and FIGO histological grading in the EC samples and the clinicopathological characteristics. Statistical significance was defined as *p* < 0.05. The analyses were performed using SPSS (ver. 24.0; IBM Corp., Armonk, NY, USA) and R (ver. 4.21; R Project for Statistical Computing, Vienna, Austria) software.

## 3. Results

### 3.1. Patient Characteristics

The clinicopathological characteristics of the patients are presented in Table 1. The mean age of the patients was 51 and more than half were premenopausal. The median tumor size and invasion depth were 3 cm and 3 mm, respectively, and twenty samples were larger than 2 cm. LVSI was positive in only 9.8% of the cases (5/51). The majority of patients were categorized as T1a (68.6%), followed by T1b (21.6%), T2 (5.9%), and a smaller proportion in T3a and T3b stages (2%). Most patients had no lymph node involvement (90.2%), with a smaller proportion exhibiting N1 (5.9%) or N2 (3.9%) involvement. Based on 2009 FIGO staging, 95.3% of the endometrial cancers were early stage. This included FIGO stage I (90.2%) and II (5.1%). With respect to FIGO histological grade, 46 (90.2%) of the patients had a lower-grade tumor (grades 1 and 2) and 5 (9.8%) had a higher-grade tumor (grade 3).

The expression pattern of GLUT-1 was defined as a mosaic if it appeared in scattered patches in the endometrial gland, central if it was distributed in the lumen of the gland, and a diffuse type if it was distributed diffusely throughout the lumen and gland (Figure 1). Because of sample loss, the GLUT-1 pattern values were only available for fifty individuals and the HIF-1α values were available for forty-nine. Twelve percent of the GLUT-1 patterns were mosaic, and 56% and 32% appeared as central and diffuse expression patterns, respectively. Thirty-two samples showed HIF-1α expression of less than 1%, which corresponded to 65.3% of the specimens, whereas forty samples (81%) showed less than 5%. Similarly, 16 specimens representing 32.7% were observed using a cut-off for HIF-1α of 1% or less in cytoplasmic-positive immune cells (IC) and 63% when the threshold was less than 5%.

### 3.2. Association of GLUT-1, HIF-1α Expression, and FIGO Histologic Grade

The association between the expression of GLUT-1 and HIF-1α and histological grade in endometrioid adenocarcinoma was analyzed based on the TMA cores. Table 2 lists the relationship between the expression patterns of GLUT-1, HIF-1α, and histologic grade. The expression of these markers was shown in both non-tumorous epithelium and endometrioid carcinoma for different FIGO histologic grades (G1, G2, and G3). The association between different grades in endometrioid adenocarcinoma was compared based on the proportion of ICs for HIF-1α. There was a significant difference in the expression of HIF-1α in ICs across all grades (*p* = 0.032), with higher expression in higher grades (G2 and G3). HIF-1α expression in tumor cells showed no significant difference between grades, unlike the ICs (*p* = 0.263). There was no significant difference in the distribution of GLUT-1 expression patterns (mosaic, central, and diffuse) across the different histological grades (*p* = 0.365).

Clinicopathological data and their association with cut-off values in the proportion of cytoplasmic HIF-1α-positive ICs are listed in Table 3. The cut-off values for the proportion of HIF-1α in the ICs at 1% and 5% with histologic grade (grades 2 and 3) were significantly associated with a larger proportion of HIF-1α in the ICs (over 1% and over 5%) and higher grade. This association was stronger at the 1% compared with the 5% (*p* = 0.048 vs. *p* = 0.036) threshold. When the GLUT-1 pattern was analyzed for the proportion of HIF-1α in the ICs based on ≤5% vs. >5%, the diffuse pattern was reported to be more significant than the central or mosaic patterns (*p* = 0.048); however, no statistical significance was observed when analyzed based on values of ≤1% vs. >1%. There were no significant differences between age, tumor size, invasion depth, LVSI, or proportion of HIF-1α in the ICs (*p* > 0.05).

## 4. Discussion

The lifetime risk of a woman developing endometrial cancer is approximately 3% and the average age at diagnosis is 61. There has been a 132% increase in the overall incidence of endometrial cancer over the last 30 years because of an increase in risk factors, particularly obesity, and an aging population [1,26]. Endometrial cancer is divided into two pathogenetic types based on metabolic, endocrine, and clinical characteristics. Type I, which accounts for 78% to 85% of the cases, occurs in younger women with unopposed estrogen, and type II occurs in older women and is an estrogen-independent tumor [27]. Most endometrial cancers are type I and are detected early, often in reproductive and perimenopausal women. Despite significant advancements in cancer management, the recurrence rate for endometrial cancer remains approximately 12% [28]. Certain histological subtypes (e.g., papillary serous carcinoma, clear cell carcinoma, and mixed carcinoma); high-grade, deep myometrial invasion; tumor extension to the cervix; spread to lymphatics or blood vessels; and extra-uterine disease are high-risk features for recurrence [29]. Traditionally, the diagnosis of endometrial carcinoma is based on histological examination. This determines both the histological type and grading of the tumor [30]. This provides prognostic information and a basis for further treatment recommendations. Some patients with endometrial cancer have similar histologic types and stages, but worse outcomes [23]. Studies are ongoing to identify independent risk factors associated with disease recurrence in endometrial cancer. This includes evaluating biochemical markers and other factors.

Recent studies have emphasized molecular subtyping based on genomic features, which involves the identification of the genetic and molecular characteristics of the tumor. This allows for a more accurate diagnosis by distinguishing between subtypes that may look similar under the microscope but exhibit different biological behaviors. Endometrial cancer was further subdivided into four different types by The Cancer Genome Atlas (TCGA) [31]. Incorporation of molecular subtyping into existing risk stratification models significantly improves prognosis over either system alone [32].

The tumor microenvironment (TME) in endometrial cancer refers to the complex cellular and non-cellular components that surround the tumor cells. These include ICs, stromal cells, the extracellular matrix, and other secreted molecules. The TME plays an important role in tumorigenesis, progression, and response to treatment [33]. The TME of many cancers exhibits low oxygen tension, particularly in solid tumors, where hypoxia is inherently linked to the pathophysiology of cancer [34]. TME in endometrial cancer can vary depending on the molecular subtype of the tumor. POLE-mutant tumors are associated with a more inflammatory TME and better prognosis, whereas p53-mutant tumors have a more immunosuppressive TME and a poorer prognosis [18,35]. Currently, biomarker-guided therapy is being used to treat endometrial cancer [35,36]; however, further studies are needed to discover biomarkers with new clinical implications beyond molecular subtypes and the TME.

Malignant tumors contain various strategies to adapt to hypoxic conditions, with significant research focusing on hypoxic markers, such as HIF-1α and GLUT-1. HIF-1α is a transcription factor that is stabilized under hypoxic conditions and is involved in the expression of GLUT-1, which is a glucose transporter [9,10]. These two substances act as facilitators of a tumor-friendly state and numerous studies have reported their association with the prognosis of various cancers. In head and neck cancers, a significant association between disease-free survival and progression-free survival has been observed, indicating a strong prognostic relationship [16,37]. The hypoxic markers, HIF-1α and GLUT-1, are associated with poor prognosis in endometrial cancer, although there are slight variations in specific parameters [18,19,20]. In this study, the histological grade of endometrial cancer exhibited a significant association with HIF-1α expression. Analysis of histological grade and HIF-1α expression in ICs using a 1% threshold for HIF-1α expression revealed that grades 2 and 3 endometrial cancer exhibited significantly higher expression levels than 1% (*p* < 0.05). The relationship with clinicopathological information (Table 3) revealed a significant difference in HIF-1α expression between a low grade (G1) and a higher grade (G2 and G3) at the 1% threshold (*p* = 0.036). Using a 5% cut-off value, the results were borderline, but still significant (*p* = 0.48). When the proportion of cells with 5% or less HIF-1α expression was present, Glut-1 exhibited a focal expression pattern that was either central or mosaic. Conversely, when HIF-1α expression exceeded 5%, GLUT-1 exhibited a diffuse expression pattern, which was significant (*p* < 0.05). This suggests that HIF-1α may induce and promote the expression of GLUT-1 [9,10]. The histologic grade of endometrial cancer has a marked impact on prognosis as reported in a recent FIGO staging update [32,38].

HIF-1α can be stabilized in an oxygen-dependent or -independent manner [39,40]. One of these ways is its stabilization in ICs, and HIF-1α is expressed in both the innate and adaptive immune cell population [21]. Recently, there has been a growing interest in the microenvironment of solid tumors, with numerous studies examining the effect of tumor microenvironment substances on cancer behavior and treatment response [41]. The percentage of HIF-1α-positive ICs in cancer can vary depending on the type of cancer. In this study, HIF-1α expression was more prominent in ICs compared with that in tumor cells, but it remains unclear whether this expression is induced by oxygen deficiency or in an oxygen-independent environment, namely, other pathogenic mechanisms of the malignant tumor. Tumors grow and progress surreptitiously by inhibiting, excluding, or ignoring the host immune system to evade recognition through tumor antigens [42]. Further studies are needed to determine which ICs are predominant and whether the expression of HIF-1α in those ICs is significantly prominent. In a study by Zhang et al., an association between HIF-1α expression and partial oxygen pressure in cancer samples was established; however, our study did not measure pO_2_ in specimens [43]. Although approximately 15 to 22 years have elapsed since sample collection, which may introduce potential errors in pO_2_ measurements, measuring pO_2_ may provide insight into the primary factors stabilizing HIF-1α.

The management of endometrial cancer should be based on an individualized assessment of the risk to the patient in an era of personalized medicine. Molecular classification has conclusively shown greater prognostic value compared with classic histology, and management based on molecular classification has yielded improved outcomes in endometrial cancer [23,30,36]. Biological markers are generally useful for the screening, diagnosis, and prognosis of a disease. There has been a great deal of research into the identification of new biological markers, which may be useful in improving the prognosis of endometrial cancer [35,36]. However, few of the biomarkers identified in endometrial cancer to date are considered specific or effective enough to be clinically useful for prognosis or predicting responses to treatment. Drugs targeting the biological mechanisms of endometrial cancer have been developed and utilized, which enhance progression-free and overall survival [44,45]. Research into the development of new therapies targeting alternative mechanisms should be pursued based on their predictive value for relapse or drug resistance. In the present study, no significant associations were evident between age, tumor size, invasion depth, and the proportion of HIF-1α in ICs (*p* > 0.05). This suggests that while HIF-1α and GLUT-1 are important markers, their expression may be modulated by factors beyond these fundamental clinicopathological parameters.

Cho et al. note that hypoxia is a common condition in the germinal center, where immune cells are highly concentrated, and discuss the role of hypoxia on immune cell function. In their study, they observed that knocking down HIF-1α in CD4+ T cells decreased the frequency of antigen-specific B cells of the germinal center and follicular T helper cells. In addition, HIF-1α was observed to promote CD40L expression while suppressing the CXCR5+ follicular regulatory population of FoxP3+ CD4+ cells. HIF-1α exerts its actions on CD4+T cells via TCR or cytokine stimulation. Taken together, these multiple findings led Cho et al. to argue that HIF-1α transcription is a component of an accessory mechanism during important humoral responses in the autoimmune system [46]. In addition, Palazon et al. suggested that CD8+ T cells utilize HIF transcription factors during adaptation to the hypoxic tumor microenvironment. They argued that a loss of HIF-1α in CD8+ T cells leads to decreased tumor infiltration and tumor cell death and altered tumor angiogenesis. They linked HIF-1α to VEGF-A expression, one of the important mechanisms for tumor angiogenesis, suggesting that the HIF-1a/VEGF-A axis is essential for tumor immunity [47]. Liu et al. interpreted HIF-1α from the perspective of T-cell exhaustion in their report. They collected data from the Chinese Glioma Genome Atlas (CGGA) database on 692 patients and investigated the association between HIF-1α and T-cell exhaustion-related genes and immune cells. They reported that high HIF-1α expression in glioma was associated with increased T-cell exhaustion-related genes and immune cell numbers. He suggested that hypoxic environments have a negative impact on antitumor immunity and that reversing the hypoxic state increases the efficacy of immunotherapy. Based on these studies, it seems that antitumor immunity increases in endometrioid carcinoma as the histologic grade increases, and immune cells with increased HIF-1α expression increase around the tumor, and it is expected that maintaining HIF-1α expression in peritumor cells will be very helpful for tumor treatment [48].

The present study was limited by a small sample size, inclusion of primarily early-stage endometrial cancers, and a lack of relapse events, which prevented a comprehensive survival analysis. In addition, the specific mechanisms underlying the effect of hypoxia-regulated protein expression on endometrial cancer warrant further study. Our focus on endometrioid histology and the exclusion of other potential influencing factors, such as adjuvant treatment, molecular classification, and comorbidities, may also limit the generalizability of the findings. Despite these limitations, this study provides valuable insight into the co-expression of HIF-1α and GLUT-1 in tissue samples, suggesting potential interactions between these hypoxia-regulated proteins. The association between HIF-1α expression in ICs and higher histologic grades highlights the importance of HIF-1α within the tumor microenvironment. Future studies should focus on elucidating the specific mechanisms through which HIF-1α in ICs influences tumor progression and its potential as a prognostic marker. In addition, larger, more comprehensive studies incorporating diverse histological types, adjuvant treatment modalities, molecular classification, and comorbidities are needed to fully understand the complex interplay between HIF-1α, GLUT-1, and other factors in endometrial cancer. Such studies may lead to the development of novel therapeutic strategies to target HIF-1α in the immune microenvironment.

## 5. Conclusions

This study provides evidence for the intricate relationship between HIF-1α and GLUT-1 expression and the clinicopathological characteristics of endometrial cancer. These findings contribute to a better understanding of the molecular underpinnings of endometrial carcinoma and underscore the potential of HIF-1α and GLUT-1 as biomarkers for tumor biology and prognosis. Further studies are warranted to examine these associations in larger cohorts and to delve into the therapeutic implications of targeting these pathways in endometrial cancer.

## Figures and Tables

**Figure 1 diagnostics-14-01735-f001:**
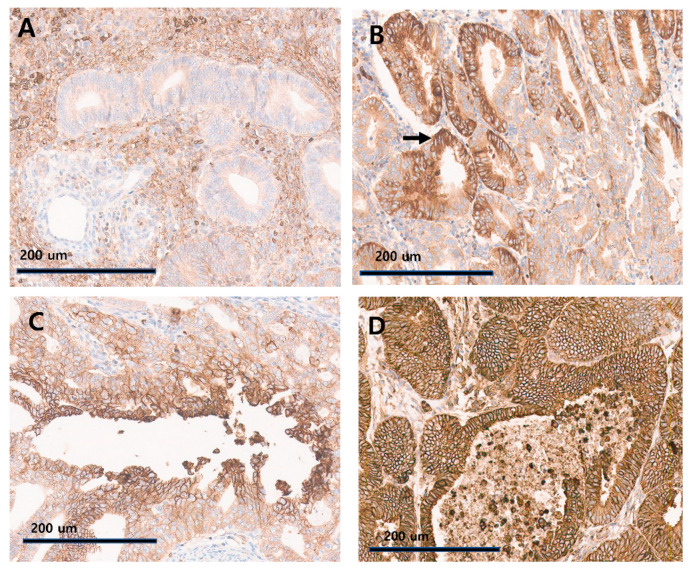
GLUT-1 expression was classified according to its distribution pattern. GLUT-1-positive cells were classified into central, mosaic, and diffuse patterns. (**A**) Non-tumorous endometrial epithelium showed no marked GLUT-1-positive cells (×200). (**B**) For the mosaic pattern, GLUT-1 strongly positive cells were distributed in patches. The black arrow indicates an area showing GLUT-1 strongly positive cells (×200). (**C**) For the central pattern, GLUT-1 strongly positive cells were distributed in the lumen of the gland (×200). (**D**) For the diffuse pattern, GLUT-1 strongly positive cells were diffusely distributed (×200).

**Figure 2 diagnostics-14-01735-f002:**
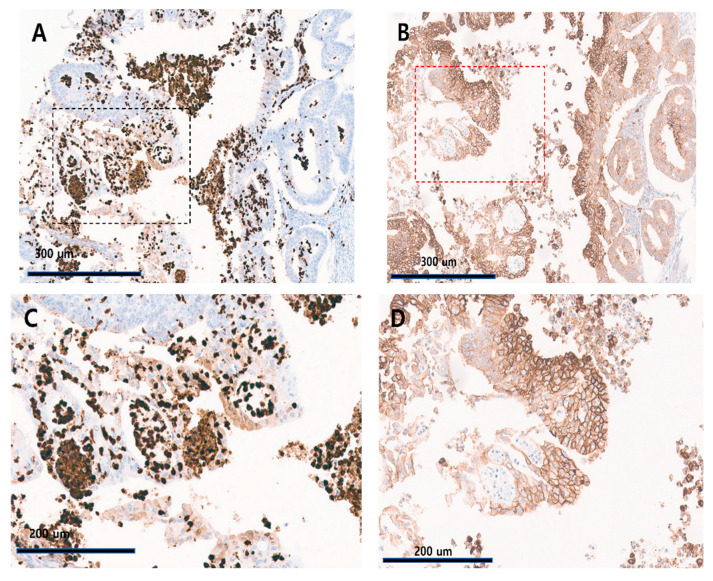
Expression of HIF-1α and GLUT-1 in tumor specimens. (**A**) A core from one of the 35 specimens exhibiting HIF-1α expression in 20% of the tumor cells and 40% of the immune cells (×100). (**B**) A core showing GLUT-1 expression with a central pattern (×100). (**C**) A magnified image of Figure 2A (black dotted rectangle, ×200). (**D**) A magnified image of Figure 2B (red dotted rectangle, ×200).

**Table 1 diagnostics-14-01735-t001:** Clinicopathological information for 51 endometrioid carcinoma patients.

Variables			Value (Median or Proportion)
Age			35~78 (51)
Tumor size (cm)			0.2~10 (3)
Invasion depth (mm)			0.1~40 (3)
LVSI, *n* (%)	Negative		46 (90.2%)
	Positive		5 (9.8%)
T stage	1a		35 (68.6%)
	1b		11 (21.6%)
	2		3 (5.9%)
	3a		1 (2%)
	3b		1 (2%)
N stage	0		46 (90.2%)
	1		3 (5.9%)
	2		2 (3.9%)
FIGO stage, *n* (%)	IA		34 (66.7%)
	IB		9 (17.7%)
	II		2 (3.9%)
	3A		1 (2.0%)
	3C		5 (9.8%)
FIGO histologic grade	G1		34 (66.6%)
	G2		12 (23.5%)
	G3		5 (9.8%)
GLUT-1 pattern *	Mosaic		6 (12.0%)
	Central		28 (56.0%)
	Diffuse		16 (32.0%)
HIF-1 α **	TC ^a^	≤1%	32 (65.3%)
		≤5%	40 (81.6%)
	IC ^b^	≤1%	16 (32.7%)
		≤5%	31 (63.3%)

LVSI: lymph vascular surface invasion; FIGO: International Federation of Gynecology and Obstetrics; GLUT-1: glucose transporter 1; HIF-1 α: hypoxia-inducible factor 1. * Because of specimen loss, values were only available for 50 patients. ** Because of specimen loss, values were only available for 49 patients. ^a^ the proportion of cytoplasmic-positive tumor cells; ^b^ the proportion of cytoplasmic-positive immune cells.

**Table 2 diagnostics-14-01735-t002:** Association of GLUT-1 and HIF-1α expression and FIGO histologic grade in EC patients.

		Non-Tumorous Epithelium	Endometrioid Carcinoma			
			G1 ^c^	G2	G3	*p*-Value
GLUT-1 pattern	Mosaic	0 *	6	0	0	0.365
	Central	0	18	6	4	
	Diffuse	0	10	5	1	
TCs ^a^ of HIF-1α	≤1%	3 **	23	5	4	0.263
	>1%	0	10	6	1	
ICs ^b^ of HIF-1α	≤1%	3	14	0	2	0.032
	>1%	0	19	11	3	

GLUT-1: glucose transporter 1; HIF-1 α: hypoxia-inducible factor 1; TCs: tumor cells; ICs: immune cells. * Four tissue microarray cores containing non-tumorous endometrial epithelium in GLUT-1 slides; ** three tissue microarray cores containing non-tumorous endometrial epithelium in HIF-1α slides; ^a^ the proportion of cytoplasmic-positive tumor cells; ^b^ the proportion of cytoplasmic-positive immune cells; ^c^ FIGO histologic grade 1.

**Table 3 diagnostics-14-01735-t003:** Association between clinicopathological factors and the proportion of cytoplasmic-positive immune cells for HIF-1α.

		The Proportion of Cytoplasmic HIF-1α-Positive Immune Cells					
		≤1%	>1%	*p*-Value	≤5%	>5%	*p*-Value
Age (years)	≤51	9	16	0.610	16	9	0.913
	>51	7	17		15	9	
Tumor size (cm)	≤3	7	15	0.910	14	8	0.961
	>3	9	18		17	10	
MI	<1/2	9	13	0.386	16	6	0.140
	≥1/2	5	13		9	9	
Histologic grade	1	14	19	0.036	24	9	0.048
	2 or 3	2	14		7	9	
GLUT-1 pattern	Central, mosaic	12	21	0.426	24	9	0.048
	Diffuse	4	12		7	9	

MI: myometrial invasion; GLUT-1: glucose transporter 1; HIF-1α: hypoxia-inducible factor 1α.

## Data Availability

The data sets that were used and/or analyzed in the current study are available from the corresponding author upon reasonable request.

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
