# Peer review of "Hypoxia-Regulated Proteins: Expression in Endometrial Cancer and Their Association with Clinicopathologic Features"

_diagnostics, 2024, doi:10.3390/diagnostics14161735_

Round 1

Reviewer 1 Report

Comments and Suggestions for Authors

Dear Authors, 

Thank you for submitting your interesting work to Diagnostics! Here are my indications/comments:

1. The introduction section is comprehensive enough. 

2. In the materials and methods section you stated the following:" The percentage of HIF-1α positive immune cells in the tissue cores was measured and recorded by two pathologists. The percentage thresholds were 1% and 5%, respectively". How did you choose these thresholds? Do you have any supporting literature/guidelines data? If yes, please refer them in this section. 

3. Materials and methods: "Correlation analyses of categorical variables were performed using the chi-squared test and Fisher’s exact test to determine the association between HIF-1α, GLUT-1 expression, and FIGO histological grading in the EC samples and the clinicopathological characteristics." Chi-squared and Fisher’s exact tests measure the association, NOT the correlation. For correlation purposes, please refer to Pearson/Spearman/Kendall analyses. 

4. Please replace the term "correlation" with "association" in the results and discussion section. You did not report any corelation coefficients. 

5. In the discussion section you should try to explain why your results indicated a higher proportion of HIF-1α positive immune cells in tumors with higher histologic grade. This finding is interesting and could open future direction of research. 

6. The conclusions are reasonable. 

Comments on the Quality of English Language

Minor English typos should be addressed. The conclusions seem to be written by Chatgpt. 

Reviewer 2 Report

Comments and Suggestions for Authors

Thank your effort for this study.

In Immunohistochemical analysis, you are saying that non-tumorous tissue is observed in 4 of 50 samples in the GLUT-1 slides. However, you have 50 cases with immunohistochemical staining for GLUT-1 in table 2. Do you think that staining pattern of non-tumorous tissues for GLUT-1 affects the final conclusion?  It may be necessary to remove out the patient with non-tumorous tissue in there paraffin blocks. There are same problem in HIF-1 group. Totally 7 cases with non-tumorous tissue in paraffin blocks may be excluded from the study.

At line 191-192, the sentence has no verb.   … Should be re-written.

At line 231,  endometrioid carcinoma should be removed out from the sentence.

At line 234, this sentence should be re-written. Statement is discussible.

Comments on the Quality of English Language

minor revision is necsesary for english language 
